# Dual Principal Component Pursuit:
# Improved Analysis and Efficient Algorithms

**Zhihui Zhu**
MINDS
Johns Hopkins University
zzhu29@jhu.edu

**Yifan Wang**
SIST
ShanghaiTech University
wangyf@shanghaitech.edu.cn

**Daniel Robinson**
AMS
Johns Hopkins University
daniel.p.robinson@jhu.edu

**Daniel Naiman**
AMS
Johns Hopkins University
daniel.naiman@jhu.edu

**Rene Vidal**
MINDS
Johns Hopkins University
rvidal@jhu.edu

**Manolis C. Tsakiris**
SIST
ShanghaiTech University
mtsakiris@shanghaitech.edu.cn

## Abstract

Recent methods for learning a linear subspace from data corrupted by outliers are based on convex $\ell_1$ and nuclear norm optimization and require the dimension of the subspace and the number of outliers to be sufficiently small [27]. In sharp contrast, the recently proposed *Dual Principal Component Pursuit (DPCP)* method [22] can provably handle subspaces of high dimension by solving a non-convex $\ell_1$ optimization problem on the sphere. However, its geometric analysis is based on quantities that are difficult to interpret and are not amenable to statistical analysis. In this paper we provide a refined geometric analysis and a new statistical analysis that show that DPCP can tolerate as many outliers as the *square* of the number of inliers, thus improving upon other provably correct robust PCA methods. We also propose a scalable *Projected Sub-Gradient Method* (DPCP-PSGM) for solving the DPCP problem and show that it achieves linear convergence even though the underlying optimization problem is non-convex and non-smooth. Experiments on road plane detection from 3D point cloud data demonstrate that DPCP-PSGM can be more efficient than the traditional RANSAC algorithm, which is one of the most popular methods for such computer vision applications.

## 1 Introduction

Fitting a linear subspace to a dataset corrupted by outliers is a fundamental problem in machine learning and statistics, primarily known as *(Robust) Principal Component Analysis (PCA)* [10, 2]. The classical formulation of PCA, dating back to Carl F. Gauss, is based on minimizing the sum of squares of the distances of all points in the dataset to the estimated linear subspace. Although this problem is non-convex, it admits a closed form solution given by the span of the top eigenvectors of the data covariance matrix. Nevertheless, it is well-known that the presence of outliers can severely affect the quality of the computed solution because the Euclidean norm is not robust to outliers.

The sensitivity of classical $\ell_2$-based PCA to outliers has been addressed by using robust maximum likelihood covariance estimators, such as the one in [25]. However, the associated optimization problems are non-convex and thus difficult to provide global optimality guarantees. Another classical approach is the exhaustive-search method of *Random Sampling And Consensus (RANSAC)* [5], which given a time budget, computes at each iteration a $d$-dimensional subspace as the span of $d$ randomly chosen points, and outputs the subspace that agrees with the largest number of points. Even though RANSAC is currently one of the most popular methods in many computer vision applications such as

Table 1: Probabilistic upper bounds for the number $M$ of tolerated outliers as a function of the number $N$ of inliers, the subspace dimension $d$, and the ambient dimension $D$, by different methods under a random Gaussian or random spherical model.

| Method | Random Gaussian Model |
|---|---|
| GGD [16] | $M \lesssim \frac{\sqrt{D(D-d)}}{d} N$ |
| REAPER [13] | $M \lesssim \frac{D}{d} N, \quad d \leq \frac{D-1}{2}$ |
| GMS [30] | $M \lesssim \frac{\sqrt{(D-d)D}}{d} N$ |
| $\ell_{2,1}$-RPCA [27] | $M \lesssim \frac{1}{d \max(1, \frac{\log(M+N)}{d})} N$ |
| TME [29] | $M < \frac{D-d}{d} N$ |
| TORP [3] | $M \lesssim \frac{1}{d \max(1, \frac{\log(M+N)}{d})^2} N$ |

| Method | Random Spherical Model |
|---|---|
| FMS [11] | $N/M \gtrsim 0$, $N \to \infty$, i.e., any ratio of outliers when[1] $N \to \infty$ |
| CoP [19] | $M \lesssim \frac{D-d^2}{d} N, \quad d < \sqrt{D}$ |
| DPCP (**this paper**) | $M \lesssim \frac{1}{dD \log^2 D} N^2$ |

multiple view geometry [9], its performance is sensitive to the choice of a thresholding parameter. Moreover, the number of required samplings may become prohibitive in cases when the number of outliers is very large and/or the subspace dimension $d$ is large and close to the dimension $D$ of the ambient space (i.e., the high relative dimension case).

As an alternative to traditional robust subspace learning methods, during the last decade ideas from compressed sensing have given rise to a new class of methods that are based on convex optimization, and admit elegant theoretical analyses and efficient algorithmic implementations. Prominent examples are based on decomposing the data matrix into low-rank and column-sparse parts [27], expressing each data point as a sparse linear combination of other data points [20, 28], and measuring the coherence of each point with every other point in the dataset [19]. The main limitation of these methods is that they are theoretically justifiable only for subspaces of low relative dimension $d/D$. However, for applications such as 3D point cloud analysis, two/three-view geometry in computer vision, and system identification, a subspace of dimension $D-1$ (high relative dimension) is sought [21, 26]. A promising direction towards handling subspaces of high relative dimension is minimizing the sum of the distances of the points to the subspace, which is a non-convex problem that REAPER [13] relaxes to a Semi-Definite Program (SDP). Even though in practice REAPER outperforms low-rank methods [27, 20, 28, 19] for subspaces of high relative dimension, its theoretical guarantees still require $d < (D-1)/2$. This is improved upon by the recent work of [16], which studies a gradient descent algorithm on the Grassmannian, and establishes convergence with high-probability to the inlier subspace for any $d/D$, as long as the number of outliers $M$ scales as $(D/d)O(N)$.

The focus of the present paper is the recently proposed *Dual Principal Component Pursuit (DPCP)* method [22, 24, 23], which seeks to learn recursively a basis for the orthogonal complement of the subspace by solving an $\ell_1$ minimization problem on the sphere. In fact, this optimization problem is precisely the underlying non-convex problem associated to REAPER and [16] for the special case $d = D - 1$. As shown in [22, 24], as long as the points are well distributed in a certain deterministic sense, any global minimizer of this non-convex problem is guaranteed to be a vector orthogonal to the subspace, regardless of the outlier/inlier ratio and the subspace dimension; a result that agrees with the earlier findings of [14]. Indeed, for synthetic data drawn from a hyperplane ($d = D - 1$), DPCP has been shown to be the only method able to correctly recover the subspace with up to 70% outliers ($D = 30$). Nevertheless, the analysis of [22, 24] involves geometric quantities that are difficult to analyze in a probabilistic setting, and consequently it has been unclear how the number $M$ of outliers that can be tolerated scales as a function of the number $N$ of inliers. Moreover, even though [22, 24] show that relaxing the non-convex problem to a sequence of linear programs (LPs) guarantees finite convergence to a vector orthogonal to the subspace, this approach is computationally expensive. Alternatively, while the *Iteratively Reweighted Least Squares* (IRLS) scheme proposed in [24, 23] is more efficient than the linear programming approach, it comes with no theoretical guarantees and scales poorly for high-dimensional data, since it involves an SVD at each iteration.

In this paper we make the following specific contributions:

- *Theory:* An improved analysis of global optimality for DPCP that replaces the cumbersome geometric quantities of [22, 24] with new quantities that are both tighter and easier to bound in probability. Specifically, employing a spherical random model suggests that DPCP can handle $M = O(\frac{1}{dD \log^2 D} N^2)$ outliers. This is in sharp contrast to existing provably correct state-of-the-art robust PCA methods, which as per Table 1 can tolerate at best $M = O(N)$ outliers.[2]

- *Algorithms:* A scalable *Projected Sub-Gradient Method* algorithm with piecewise geometrically diminishing step sizes (DPCP-PSGM), which is proven to solve the non-convex DPCP problem with linear convergence and using only matrix-vector multiplications. This is in contrast to classic results in the literature on the PSGM, which usually requires the problem to be convex in order to establish sub-linear convergence [1]. DPCP-PSGM is orders of magnitude faster than the LP-based and IRLS schemes proposed in [24], which allows us to extend the size of the datasets that we can handle from $10^3$ to $10^6$ data points.

- *Experiments:* Experiments on road plane detection from 3D point cloud data using the KITTI dataset [6], which is an important computer vision task in autonomous car driving systems, show that for the same computational budget DPCP-PSGM outperforms RANSAC, which is one of the most popular methods for such computer vision applications.

## 2   Global Optimality Analysis for Dual Principal Component Pursuit

**Review of DPCP**   Given a unit $\ell_2$-norm dataset $\widetilde{\boldsymbol{\mathcal{X}}} = [\boldsymbol{\mathcal{X}} \; \boldsymbol{\mathcal{O}}]\boldsymbol{\Gamma} \in \mathbb{R}^{D \times L}$, where $\boldsymbol{\mathcal{X}} \in \mathbb{R}^{D \times N}$ are inlier points spanning a $d$-dimensional subspace $\mathcal{S}$ of $\mathbb{R}^D$, $\boldsymbol{\mathcal{O}}$ are outlier points having no linear structure, and $\boldsymbol{\Gamma}$ is an unknown permutation, the goal of robust PCA is to recover the inlier space $\mathcal{S}$ or equivalently to cluster the points into inliers and outliers. Towards that end, the main idea of Dual Principal Component Pursuit (DPCP) [22, 24] is to first compute a hyperplane $\mathcal{H}_1$ that contains all the inliers $\boldsymbol{\mathcal{X}}$. Such a hyperplane can be used to discard a potentially very large number of outliers, after which a method such as RANSAC may successfully be applied to the reduced dataset [3]. Alternatively, if $d$ is known, then one may proceed to recover $\mathcal{S}$ as the intersection of $D - d$ orthogonal hyperplanes that contain $\boldsymbol{\mathcal{X}}$. In any case, DPCP computes a normal vector $\boldsymbol{b}_1$ to the first hyperplane $\mathcal{H}_1$ as follows:

$$\min_{\boldsymbol{b} \in \mathbb{R}^D} \|\widetilde{\boldsymbol{\mathcal{X}}}^\top \boldsymbol{b}\|_0 \; \text{s.t.} \; \boldsymbol{b} \neq 0. \tag{1}$$

Notice that the function $\|\widetilde{\boldsymbol{\mathcal{X}}}^\top \boldsymbol{b}\|_0$ being minimized simply counts how many points in the dataset are not contained in the hyperplane with normal vector $\boldsymbol{b}$. Assuming that there are at least $d + 1$ inliers and at least $D - d$ outliers (this is to avoid degenerate situations), and that all points are in general position[4], then every solution $\boldsymbol{b}^*$ to (1) must correspond to a hyperplane that contains $\boldsymbol{\mathcal{X}}$, and hence $\boldsymbol{b}^*$ is orthogonal to $\mathcal{S}$. Since (1) is computationally intractable, it is reasonable to replace it by[5]

$$\min_{\boldsymbol{b} \in \mathbb{R}^D} f(\boldsymbol{b}) := \|\widetilde{\boldsymbol{\mathcal{X}}}^\top \boldsymbol{b}\|_1 \; \text{s.t.} \; \|\boldsymbol{b}\|_2 = 1. \tag{2}$$

Although problem (2) is non-convex (because of the constraint) and non-smooth (because of the $\ell_1$ norm), the work of [22, 24] established conditions suggesting that if the outliers are well distributed on the unit sphere and the inliers are well distributed on the intersection of the unit sphere with the subspace $\mathcal{S}$, then global minimizers of (2) are orthogonal to $\mathcal{S}$. Nevertheless, these conditions are deterministic in nature and difficult to interpret. In this section, we give improved global optimality conditions that are i) tighter, ii) easier to interpret and iii) amenable to a probabilistic analysis.

**Geometry of the critical points**   The heart of our analysis lies in a tight geometric characterization of the critical points of (2) (see Lemma 1 below). Before stating the result, we need to introduce some further notation and definitions. Letting $\mathcal{P}_\mathcal{S}$ be the orthogonal projection onto $\mathcal{S}$, we define the *principal angle* of $\boldsymbol{b}$ from $\mathcal{S}$ as $\phi \in [0, \frac{\pi}{2}]$ such that $\cos(\phi) = \|\mathcal{P}_\mathcal{S}(\boldsymbol{b})\|_2 / \|\boldsymbol{b}\|_2$. Since we will

consider the first-order optimality conditions of (2), we naturally need to compute the sub-differential of the objective function in (2). Towards that end, we denote the sign function by $\text{sign}(a) = a/|a|$ when $a \neq 0$, and $\text{sign}(a) = 0$ when $a = 0$. We also require the sub-differential Sgn of the absolute value function $|a|$ defined as $\text{Sgn}(a) = \text{sign}(a)$ when $a \neq 0$, and $\text{Sgn}(a) = [-1, 1]$ when $a = 0$. We use $\text{sign}(\boldsymbol{a})$ to indicate that we apply the sign function element-wise to the vector $\boldsymbol{a}$ and similarly for Sgn. Next, global minimizers of (2) are critical points in the following sense:

**Definition 1.** *A vector $\boldsymbol{b} \in \mathbb{S}^{D-1}$ is called a critical point of (2) if there exists $\boldsymbol{d}' \in \partial f(\boldsymbol{b})$ such that the Riemannian gradient $\boldsymbol{d} := (\mathbf{I} - \boldsymbol{b}\boldsymbol{b}^\top)\boldsymbol{d}' = \boldsymbol{0}$, where $\partial f(\boldsymbol{b}) = \widetilde{\boldsymbol{\mathcal{X}}} \, \text{Sgn}(\widetilde{\boldsymbol{\mathcal{X}}}^\top \boldsymbol{b})$ is the sub-differential of $f$ at $\boldsymbol{b}$.*

We now illustrate the key idea behind characterizing the geometry of the critical points. Let $\boldsymbol{b}$ be a critical point that is not orthogonal to $\mathcal{S}$. Then, under general position assumptions on the data, $\boldsymbol{b}$ can be orthogonal to $K \leq D - 1$ columns of $\widetilde{\boldsymbol{\mathcal{X}}}$, of which at most $d - 1$ can be inliers (otherwise $\boldsymbol{b} \perp \mathcal{S}$). It follows that any Riemannian sub-gradient evaluated at $\boldsymbol{b}$ has the form

$$\boldsymbol{d} = (\mathbf{I} - \boldsymbol{b}\boldsymbol{b}^\top)\boldsymbol{\mathcal{O}} \, \text{sign}(\boldsymbol{\mathcal{O}}^\top \boldsymbol{b}) + (\mathbf{I} - \boldsymbol{b}\boldsymbol{b}^\top)\boldsymbol{\mathcal{X}} \, \text{sign}(\boldsymbol{\mathcal{X}}^\top \boldsymbol{b}) + \boldsymbol{\xi}, \tag{3}$$

where $\boldsymbol{\xi} = \sum_{i=1}^{K} \alpha_{j_i} \tilde{\boldsymbol{x}}_{j_i}$ with $\tilde{\boldsymbol{x}}_{j_1}, \ldots, \tilde{\boldsymbol{x}}_{j_K}$ the columns of $\widetilde{\boldsymbol{\mathcal{X}}}$ orthogonal to $\boldsymbol{b}$ and $\alpha_{j_1}, \ldots, \alpha_{j_K} \in [-1, 1]$. Note that $\|\boldsymbol{\xi}\|_2 < D$. Since $\boldsymbol{b}$ is a critical point, Definition 1 implies a choice of $\alpha_{j_i}$ so that $\boldsymbol{d} = \boldsymbol{0}$. Define $\boldsymbol{b} = \cos(\phi)\boldsymbol{s} + \sin(\phi)\boldsymbol{n}$, where $\phi$ is the principal angle of $\boldsymbol{b}$ from $\mathcal{S}$, and $\boldsymbol{s} = \mathcal{P}_{\mathcal{S}}(\boldsymbol{b})/\|\mathcal{P}_{\mathcal{S}}(\boldsymbol{b})\|_2$ and $\boldsymbol{n} = \mathcal{P}_{\mathcal{S}^\perp}(\boldsymbol{b})/\|\mathcal{P}_{\mathcal{S}^\perp}(\boldsymbol{b})\|_2$ are the orthonormal projections of $\boldsymbol{b}$ onto $\mathcal{S}$ and $\mathcal{S}^\perp$, respectively. Defining $\boldsymbol{g} = -\sin(\phi)\boldsymbol{s} + \cos(\phi)\boldsymbol{n}$ and noting that $\boldsymbol{g} \perp \boldsymbol{b}$, it follows that

$$0 = \boldsymbol{g}^\top \boldsymbol{\mathcal{O}} \, \text{sign}(\boldsymbol{\mathcal{O}}^\top \boldsymbol{b}) - \sin(\phi) \left\| \boldsymbol{\mathcal{X}}^\top \boldsymbol{s} \right\|_1 + \boldsymbol{g}^\top \boldsymbol{\xi}, \tag{4}$$

which in particular implies that

$$\sin(\phi) \leq \left( \left| \boldsymbol{g}^\top \boldsymbol{\mathcal{O}} \, \text{sign}(\boldsymbol{\mathcal{O}}^\top \boldsymbol{b}) \right| + D \right) / \left\| \boldsymbol{\mathcal{X}}^\top \boldsymbol{s} \right\|_1. \tag{5}$$

Thus, we obtain Lemma 1 after defining

$$\eta_{\boldsymbol{\mathcal{O}}} := \frac{1}{M} \max_{\boldsymbol{g}, \boldsymbol{b} \in \mathbb{S}^{D-1}, \boldsymbol{g} \perp \boldsymbol{b}} \left| \boldsymbol{g}^\top \boldsymbol{\mathcal{O}} \, \text{sign}(\boldsymbol{\mathcal{O}}^\top \boldsymbol{b}) \right| \quad \text{and} \quad c_{\boldsymbol{\mathcal{X}}, \min} := \frac{1}{N} \min_{\boldsymbol{b} \in \mathcal{S} \cap \mathbb{S}^{D-1}} \|\boldsymbol{\mathcal{X}}^\top \boldsymbol{b}\|_1. \tag{6}$$

**Lemma 1.** *Any critical point $\boldsymbol{b}$ of (2) must either be a normal vector of $\mathcal{S}$, or have a principal angle $\phi$ from $\mathcal{S}$ smaller than or equal to $\arcsin\left(M\overline{\eta}_{\boldsymbol{\mathcal{O}}}/Nc_{\boldsymbol{\mathcal{X}}, \min}\right)$, where $\overline{\eta}_{\boldsymbol{\mathcal{O}}} := \eta_{\boldsymbol{\mathcal{O}}} + D/M$.*

Towards interpreting Lemma 1, we first give some insight into the quantities $\eta_{\boldsymbol{\mathcal{O}}}$ and $c_{\boldsymbol{\mathcal{X}}, \min}$. First, we claim that $\eta_{\boldsymbol{\mathcal{O}}}$ reflects how well distributed the outliers are, with smaller values corresponding to more uniform distributions. This can be seen by noting that as $M \to \infty$ and assuming that $\boldsymbol{\mathcal{O}}$ remains well distributed, the quantity $\frac{1}{M}\boldsymbol{\mathcal{O}} \, \text{sign}\left(\boldsymbol{\mathcal{O}}^\top \boldsymbol{b}\right)$ tends to the quantity $c_D \boldsymbol{b}$, where $c_D$ is the average height of the unit hemi-sphere of $\mathbb{R}^D$ [22, 24]. Since $\boldsymbol{g} \perp \boldsymbol{b}$, in the limit $\eta_{\boldsymbol{\mathcal{O}}} \to 0$. Second, the quantity $c_{\boldsymbol{\mathcal{X}}, \min}$ is the same as the *permeance statistic* defined in [13], and for well-distributed inliers is bounded away from small values, since there is no single direction in $\mathcal{S}$ sufficiently orthogonal to $\boldsymbol{\mathcal{X}}$. We thus see that according to Lemma 1, any critical point of (2) is either orthogonal to the inlier subspace $\mathcal{S}$, or very close to $\mathcal{S}$, with its principal angle $\phi$ from $\mathcal{S}$ being smaller for well distributed points and smaller outlier to inlier ratios $M/N$. Interestingly, Lemma 1 suggests that any algorithm can be utilized to find a normal vector to $\mathcal{S}$ as long as the algorithm is guaranteed to find a critical point of (2) and this critical point is sufficiently far from the subspace $\mathcal{S}$, i.e., it has principal angle larger than $\arcsin\left(M\overline{\eta}_{\boldsymbol{\mathcal{O}}}/Nc_{\boldsymbol{\mathcal{X}}, \min}\right)$. We will utilize this crucial observation in the next section to derive guarantees for convergence to the global optimum for a new scalable algorithm.

**Global optimality** In order to characterize the global solutions of (2), we define quantities similar to $c_{\boldsymbol{\mathcal{X}}, \min}$ but associated with the outliers, namely

$$c_{\boldsymbol{\mathcal{O}}, \min} := \frac{1}{M} \min_{\boldsymbol{b} \in \mathbb{S}^{D-1}} \|\boldsymbol{\mathcal{O}}^\top \boldsymbol{b}\|_1 \quad \text{and} \quad c_{\boldsymbol{\mathcal{O}}, \max} := \frac{1}{M} \max_{\boldsymbol{b} \in \mathbb{S}^{D-1}} \|\boldsymbol{\mathcal{O}}^\top \boldsymbol{b}\|_1. \tag{7}$$

The next theorem, whose proof relies on Lemma 1, provides new deterministic conditions under which any global solution to (2) must be a normal vector to $\mathcal{S}$.

**Theorem 1.** *Any global solution $\boldsymbol{b}^\star$ to (2) must be orthogonal to the inlier subspace $\mathcal{S}$ as long as*

$$\frac{M}{N} \cdot \frac{\sqrt{\overline{\eta}_{\boldsymbol{\mathcal{O}}}^2 + (c_{\boldsymbol{\mathcal{O}}, \max} - c_{\boldsymbol{\mathcal{O}}, \min})^2}}{c_{\boldsymbol{\mathcal{X}}, \min}} < 1. \tag{8}$$

Towards interpreting Theorem 1, recall that for well distributed inliers and outliers $\overline{\eta}_{\mathcal{O}}$ is small, while the permeance statistics $c_{\mathcal{O},\max}$, $c_{\mathcal{O},\min}$ are bounded away from small values. Now, the quantity $c_{\mathcal{O},\max}$, thought of as a *dual* permeance statistic, is bounded away from large values for the reason that there is not a single direction in $\mathbb{R}^D$ that can sufficiently capture the distribution of $\mathcal{O}$. In fact, as $M$ increases the two quantities $c_{\mathcal{O},\max}$, $c_{\mathcal{O},\min}$ tend to each other and their difference goes to zero as $M \to \infty$. With these insights, Theorem 1 implies that regardless of the outlier/inlier ratio $M/N$, as we have more and more inliers and outliers while keeping $D$ and $M/N$ fixed, and assuming the points are well-distributed, condition (8) will eventually be satisfied and any global minimizer must be orthogonal to the inlier subspace $\mathcal{S}$.

A similar condition to (8) is given in [22, Theorem 2]. Although the proofs of the two theorems share some common elements, [22, Theorem 2] is derived by establishing discrepancy bounds between (2) and a *continuous* analogue of (2), and involves quantities difficult to handle such as *spherical cap discrepancies* and circumradii of *zonotopes*. In addition, as shown in Figure 1, a numerical comparison of the conditions of the two theorems reveals that condition (8) is much tighter. We attribute this to the quantities in our new analysis better representing the function $\|\widetilde{\mathcal{X}}^{\top}\boldsymbol{b}\|_1$ being minimized, namely $c_{\mathcal{X},\min}$, $c_{\mathcal{O},\min}$, $c_{\mathcal{O},\max}$, and $\overline{\eta}_{\mathcal{O}}$, when compared to the quantities used in the analysis of [22, 24]. Moreover, our quantities are easier to bound under a probabilistic model, thus leading to the following characterization of the number of outliers that may be tolerated.

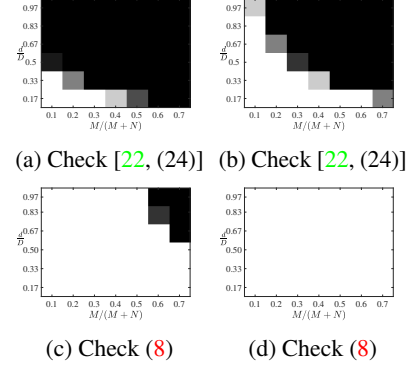

(a) Check [22, (24)]   (b) Check [22, (24)]

(c) Check (8)   (d) Check (8)

Figure 1: Check whether the condition (8) and a similar condition in [22, Theorem 2] are satisfied (white) or not (black) for a fixed number $N$ of inliers while varying the outlier ratio $M/(M+N)$ and the subspace relative dimension $d/D$: (a)-(c), $N = 500$; (b)-(d), $N = 1000$.

**Theorem 2.** *Consider a random spherical model where the columns of $\mathcal{O}$ are drawn uniformly from the sphere $\mathbb{S}^{D-1}$ and the columns of $\mathcal{X}$ are drawn uniformly from $\mathbb{S}^{D-1} \cap \mathcal{S}$, where $\mathcal{S}$ is a subspace of dimension $d < D$. Fix any $t < 2(c_d\sqrt{N} - 2)$. Then with probability at least $1 - 6e^{-t^2/2}$, any global solution of (2) is orthogonal to $\mathcal{S}$ as long as*

$$(4+t)^2 M + C_0 \left(\sqrt{D}\log D + t\right)^2 M \le \left(\sqrt{\frac{2}{\pi d}}N - (2+t/2)\sqrt{N}\right)^2, \tag{9}$$

*where $C_0$ is a universal constant that is independent of $N, M, D, d$ and $t$.*

Interestingly, Theorem 2 suggests that DPCP can roughly tolerate $M = O(\frac{1}{dD\log^2 D}N^2)$ outliers. We believe this makes DPCP the first method that is able to tolerate $O(N^2)$ outliers when $d$ and $D$ are fixed, since as per Table 1 current provably correct state-of-the-art methods can handle at best $M = O(N)$. For example, REAPER [13] requires $M \le O(\frac{D}{d}N)$. On the other hand, our bound is a decreasing function of $D$, which is an artifact of the proof technique used; we conjecture that this can be mended by a more sophisticated analysis of the term $\overline{\eta}_{\mathcal{O}}$.

Finally, our choice to use a spherical random model as opposed to a Gaussian model is a technical one: the analysis is more difficult when the functions are both non-Lipschitz and unbounded. That being said, we believe that this choice does not impose any practical limitations, since one can always normalize the data without changing the angles of the inliers/outliers to the linear subspace.

## 3   A Scalable Algorithm for Dual Principal Component Pursuit

Note that the DPCP problem (2) involves a convex objective function and a non-convex feasible region, which nevertheless is easy to project onto. This structure was exploited in [18, 22], where in the second case the authors proposed an *Alternating Linearization and Projection (ALP)* method that solves a sequence of linear programs (LP) with a linearization of the non-convex constraint and then projection onto the sphere.[6] Although efficient LP solvers (such as Gurobi [8]) may be used to solve each LP, these methods do not scale well with the problem size (i.e., $D, N$ and

$M$). Inspired by Lemma 1, which states that any critical point that has principal angle larger than $\arcsin\left(M\overline{\eta}_{\mathcal{O}}/Nc_{\boldsymbol{\mathcal{X}},\min}\right)$ must be a normal vector of $\mathcal{S}$, we now consider solving (2) with a first-order method, specifically Projected Sub-Gradient Method (DPCP-PSGM), which is stated in Algorithm 1.

---

**Algorithm 1** (DPCP-PSGM) Projected Sub-gradient Method for Solving (2)

---

**Input:** data $\widetilde{\boldsymbol{\mathcal{X}}} \in \mathbb{R}^{D \times L}$ and initial step size $\mu_0$;
**Initialization:** set $\widehat{\boldsymbol{b}}_0 = \arg\min_{\boldsymbol{b}} \|\widetilde{\boldsymbol{\mathcal{X}}}^\top \boldsymbol{b}\|_2$, s.t. $\boldsymbol{b} \in \mathbb{S}^{D-1}$;
 1: **for** $k = 1, 2, \dots$ **do**
 2:     update the step size $\mu_k$ according to a certain rule;
 3:     $\boldsymbol{b}_k = \widehat{\boldsymbol{b}}_{k-1} - \mu_k \widetilde{\boldsymbol{\mathcal{X}}} \operatorname{sign}(\widetilde{\boldsymbol{\mathcal{X}}}^\top \widehat{\boldsymbol{b}}_{k-1})$; $\widehat{\boldsymbol{b}}_k = \mathcal{P}_{\mathbb{S}^{D-1}}(\boldsymbol{b}_k) = \boldsymbol{b}_k/\|\boldsymbol{b}_k\|$;
 4: **end for**

---

Unlike projected gradient descent for smooth problems, the choice of step size for PSGM is more complicated since a constant step size in general can not guarantee the convergence of PSGM even to a critical point, though such a choice is often used in practice. For the purpose of illustration, consider a simple example $h(x) = |x|$ without any constraint, and suppose that $\mu_k = 0.08$ for all $k$ and that an initialization of $x_0 = 0.1$ is used. Then, the iterates $\{x_k\}$ will jump between two points $0.02$ and $-0.06$ and never converge to the global minimum $0$. Thus, a widely adopted strategy is to use diminishing step sizes, including those that are not summable (such as $\mu_k = O(1/k)$ or $\mu_k = O(1/\sqrt{k})$) [1], or geometrically diminishing (such as $\mu_k = O(\rho^k), \rho < 1$) [7, 4, 15]. However, for such choices, most of the literature establishes convergence guarantees for PSGM in the context of convex feasible regions [1, 7, 4], and thus can not be directly applied to Algorithm 1.

For the rest of this section, it is more convenient to use the principal angle $\theta \in [0, \frac{\pi}{2}]$ between $\boldsymbol{b}$ and the orthogonal subspace $\mathcal{S}^\perp$; thus $\boldsymbol{b}$ is a normal vector of $\mathcal{S}$ if and only if $\theta = 0$. We also need a quantity similar to $\eta_{\mathcal{O}}$ that quantifies how well the inliers are distributed within the subspace $\mathcal{S}$:

$$\eta_{\boldsymbol{\mathcal{X}}} := \frac{1}{N} \max_{\boldsymbol{g}, \boldsymbol{b} \in \mathcal{S} \cap \mathbb{S}^{D-1}, \boldsymbol{g} \perp \boldsymbol{b}} \left| \boldsymbol{g}^\top \boldsymbol{\mathcal{X}} \operatorname{sign}(\boldsymbol{\mathcal{X}}^\top \boldsymbol{b}) \right|.$$

Our next result provides performance guarantees for Algorithm 1 for various choices of step sizes ranging from constant to geometrically diminishing step sizes, the latter one giving an *R-linear convergence* of the sequence of principal angles to zero.

**Theorem 3** (Convergence guarantee for PSGM). *Let $\{\widehat{\boldsymbol{b}}_k\}$ be the sequence generated by Algorithm 1 with initialization $\widehat{\boldsymbol{b}}_0$, whose principal angle $\theta_0$ to $\mathcal{S}^\perp$ is assumed to satisfy*

$$\theta_0 < \arctan\left(\left(Nc_{\boldsymbol{\mathcal{X}},\min}\right)/\left(N\eta_{\boldsymbol{\mathcal{X}}} + M\eta_{\mathcal{O}}\right)\right). \tag{10}$$

*Let $\mu' := \frac{1}{4 \cdot \max\{Nc_{\boldsymbol{\mathcal{X}},\min}, Mc_{\mathcal{O},\max}\}}$. Assuming that $Nc_{\boldsymbol{\mathcal{X}},\min} \geq N\eta_{\boldsymbol{\mathcal{X}}} + M\eta_{\mathcal{O}}$, the angle $\theta_k$ between $\widehat{\boldsymbol{b}}_k$ and $\mathcal{S}^\perp$ satisfies the following properties in accordance with various choices of step sizes.*

*(i) (constant step size) With $\mu_k = \mu \leq \mu', \forall k \geq 0$, we have*

$$\theta_k \leq \begin{cases} \max\{\theta_0, \theta^\diamond(\mu)\}, & k < K^\diamond(\mu), \\ \theta^\diamond(\mu), & k \geq K^\diamond(\mu), \end{cases} \tag{11}$$

*where $K^\diamond(\mu) := \frac{\tan(\theta_0)}{\mu(Nc_{\boldsymbol{\mathcal{X}},\min} - \max\{1, \tan(\theta_0)\}(N\eta_{\boldsymbol{\mathcal{X}}} + M\eta_{\mathcal{O}}))}$ and $\theta^\diamond(\mu) := \arctan\left(\frac{\mu}{\sqrt{2}\mu'}\right)$.*
*(ii) (diminishing step size) With $\mu_k \leq \mu', \mu_k \to 0, \sum_{k=1}^\infty \mu_k = \infty$, we have $\theta_k \to 0$.*
*(iii) (diminishing step size of $O(1/k)$) With $\mu_0 \leq \mu', \mu_k = \frac{\mu_0}{k}, \forall k \geq 1$, we have $\tan(\theta_k) = O(\frac{1}{k})$.*
*(iv) (piecewise geometrically diminishing step size) With $\mu_0 \leq \mu'$ and*

$$\mu_k = \begin{cases} \mu_0, & k < K_0, \\ \mu_0 \beta^{\lfloor (k-K_0)/K \rfloor + 1}, & k \geq K_0, \end{cases} \tag{12}$$

*where $\beta \in (0, 1)$, $\lfloor \cdot \rfloor$ is the floor function, and $K_0, K \in \mathbb{N}$ are chosen such that*

$$K_0 \geq K^\diamond(\mu_0) \text{ and } K \geq \left(\sqrt{2}\beta\mu'\left(Nc_{\boldsymbol{\mathcal{X}},\min} - (N\eta_{\boldsymbol{\mathcal{X}}} + M\eta_{\mathcal{O}})\right)\right)^{-1} \tag{13}$$

*with $K^\diamond(\mu)$ defined right after (11), we have*

$$\tan(\theta_k) \le \begin{cases} \max\{\tan(\theta_0), \frac{\mu_0}{\sqrt{2}\mu'}\}, & k < K_0, \\ \frac{\mu_0}{\sqrt{2}\mu'}\beta^{\lfloor(k-K_0)/K\rfloor}, & k \ge K_0. \end{cases} \quad (14)$$

First note that with the choice of constant step size $\mu$, although PSGM is not guaranteed to find a normal vector, (11) ensures that after $K^\diamond(\mu)$ iterations, $\widehat{\boldsymbol{b}}_k$ is close to $\mathcal{S}^\perp$ in the sense that $\theta_k \le \theta^\diamond(\mu)$, which can be much smaller than $\theta_0$ for a sufficiently small $\mu$. The expressions for $K^\diamond(\mu)$ and $\theta^\diamond(\mu)$ indicate that there is a tradeoff in selecting the step size $\mu$. By choosing a larger step size $\mu$, we have a smaller $K^\diamond(\mu)$ but a larger upper bound $\theta^\diamond(\mu)$. We can balance this tradeoff according to the requirements of specific applications. For example, in applications where the accuracy of $\theta$ (to zero) is not as important as the convergence speed, it is appropriate to choose a larger step size. An alternative and more efficient way to balance this tradeoff is to change the step size as the iterations proceed. For the classical diminishing step sizes that are not summable, Theorem 3(ii) guarantees convergence of $\theta_k$ to zero (i.e., all limit points of the sequence of iterates $\{\widehat{\boldsymbol{b}}_k\}$ are normal vectors), though the convergence rate depends on the specific choice of step size. For example, Theorem 3(iii) guarantees a sub-linear convergence of $\tan(\theta_k)$ for step size diminishing at the rate of $1/k$.

The approach of piecewise geometrically diminishing step size (see Theorem 3(iv)) takes advantage of the tradeoff in Theorem 3(i) by first using a relatively large initial step size $\mu_0$ so that $K^\diamond(\mu_0)$ is small (although $\theta^\diamond(\mu_0)$ is large), and then decreasing the step size in a piecewise fashion. As illustrated in Figure 2, with such a piecewise geometrically diminishing step size, (14) establishes a piecewise geometrically decaying bound for the principal angles. Note that the curve $\tan(\theta_k)$ is not monotone because, as noted earlier, PSGM is not a descent method. Perhaps the most surprising aspect in Theorem 3(iv) is that with the diminishing step size (12), we obtain a $K$-step $R$-linear convergence rate for $\tan(\theta_k)$. This linear convergence rate relies on both the choice of the step size and certain beneficial geometric structure in the problem. As characterized by Lemma 1, one such structure is

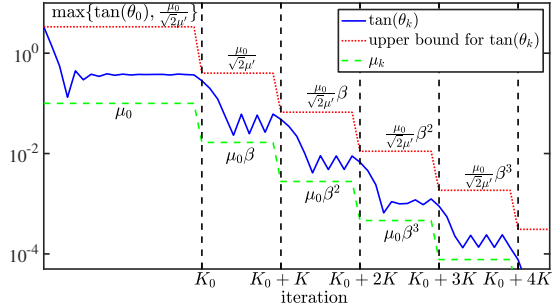

Figure 2: Illustration of Theorem 3(iv): $\theta_k$ is the principal angle between $\boldsymbol{b}_k$ and $\mathcal{S}^\perp$ generated by the PSGM Algorithm 1 with piecewise geometrically diminishing step size. The red dotted line represents the upper bound on $\tan(\theta_k)$ given by (14), while the green dashed line indicates the choice of the step size (12).

that all critical points in a neighborhood of $\mathcal{S}^\perp$ are global solutions. Aside from this, other properties (e.g., the negative direction of the Riemannian subgradient points toward $\mathcal{S}^\perp$) are used to show the decaying rate of the principal angle. This is different from the recent work [4] in which linear convergence for PSGM is obtained for sharp and weakly convex objective functions and convex constraint sets. Thus, we believe the choice of piecewise geometrically diminishing step size is of independent interest and can be useful for other nonsmooth problems.[7]

## 4 Experiments on Synthetic Data and Real 3D Point Cloud Road Data

**Synthetic Data** We first use synthetic data to verify the proposed PSGM algorithm. We fix $D = 30$, randomly sample a subspace $\mathcal{S}$ of dimension $d = 29$, and uniformly at random sample $N = 500$ inliers and $M = 1167$ outliers (so that the outlier ratio $M/(M + N) = 0.7$) with unit $\ell_2$-norm. Inspired by the Piecewise Geometrically Diminishing (PGD) step size, we also use a modified backtracking line search (MBLS) that always uses the previous step size as an initialization for finding the current one within a backtracking line search [17, Section 3.1] strategy, which dramatically reduces the computational time compared with a standard backtracking line search. The

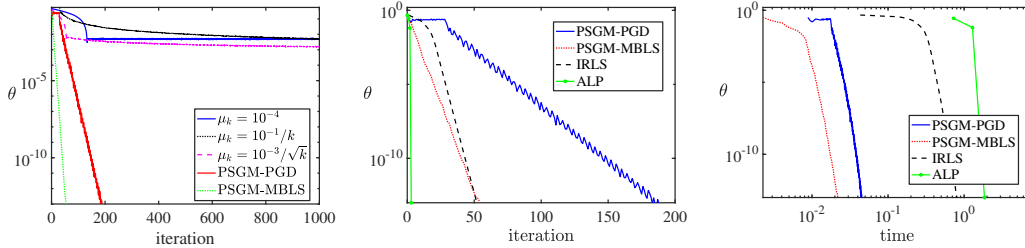

Figure 3: (L) Convergence of PSGM for different step sizes. Comparison of PSGM with ALP and IRLS in [24] in terms of (M) iterations and (R) computing time. Here $D = 30$ and $d = 29$, $N = 500$, $\frac{M}{M+N} = 0.7$.

corresponding algorithm is denoted by PSGM-MBLS. (This variant does not have any convergence guarantee for nonsmooth problems but performed well in practice.) We set $K_0 = 30$, $K = 4$ and $\beta = 1/2$ for the PGD step size with initial step size obtained by one iteration of a backtracking line search and denote the corresponding algorithm by PSGM-PGD. We define $\widehat{\boldsymbol{b}}_0$ to be the bottom eigenvector of $\widetilde{\boldsymbol{\mathcal{X}}}\widetilde{\boldsymbol{\mathcal{X}}}^\top$, which has been demonstrated to be effective in practice [24].

Figure 3(L) displays the convergence of the PSGM (Algorithm 1) with different choices of step sizes. We observe linear convergence for both PSGM-PGD and PSGM-MBLS, which converge much faster than PSGM with constant step size or classical diminishing step size. In Figure 3(M)/(R) we compare PSGM algorithms with the ALP and IRLS algorithm (referred to as DPCP-LP and DPCP-IRLS, respectively, in [24]). First observe that, as expected, although ALP finds a normal vector in few iterations, it has the highest time complexity because it solves an LP during each iteration. Figure 3(R) indicates that one iteration of ALP consumes more time than the whole procedure for PSGM. We also note that aside from the theoretical guarantee for PSGM-PGD, it also converges faster than IRLS (in terms of computing time), the latter lacking a convergence guarantee. Finally, Figure 4 illustrates Theorem 2, using the same setup, by showing the principal angle from $\mathcal{S}^\perp$ of the solution to the DPCP problem computed by the PSGM-MBLS algorithm: the phase transition is indeed quadratic, indicating that DPCP can tolerate as many as $O(N^2)$ outliers as predicted by Theorem 2.

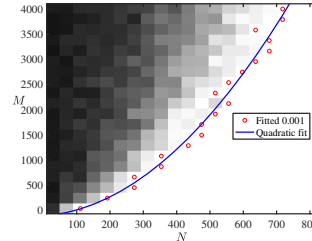

Figure 4: The principal angle $\theta$ between the solution to the DPCP problem (2) and $\mathcal{S}^\perp$: black corresponds to $\frac{\pi}{2}$ and white corresponds to 0. Here $D = 30$ and $d = 29$. For each $M$, we find the smallest $N$ (red dots) such that $\theta \leq 0.001$. The blue quadratic curve indicates the least-squares fit to these points.

**Experiments on real 3D point cloud road data**  We compare DPCP-PSGM (with a modified backtracking line search) with RANSAC [5], $\ell_{2,1}$-RPCA [27] and REAPER [13] on the road detection challenge[8] of the KITTI dataset [6], recorded from a moving platform while driving in and around Karlsruhe, Germany. This dataset consists of image data together with corresponding 3D points collected by a rotating 3D laser scanner. In this experiment we use only the 360° 3D point clouds with the objective of determining the 3D points that lie on the road plane (inliers) and those off that plane (outliers). Typically, each 3D point cloud is on the order of 100,000 points including about 50% outliers. Using homogeneous coordinates this can be cast as a robust hyperplane learning problem in $\mathbb{R}^4$. Since the dataset is not annotated for that purpose, we manually annotated a few frames (e.g., see the left column of Fig. 5). Since DPCP-PSGM is the fastest method (on average converging in about 100 milliseconds for each frame on a 6 core 6 thread Intel (R) i5-8400 machine), we set the time budget for all methods equal to the running time of DPCP-PSGM. For RANSAC we also compare with 10 and 100 times that time budget. Since $\ell_{2,1}$-RPCA does not directly return a subspace model, we extract the normal vector via SVD on the low-rank matrix returned by that method. Table 2 reports the area under the Receiver Operator Curve (ROC), the latter obtained by thresholding the distances of the points to the hyperplane estimated by each method, using a suitable range of different thresholds[9]. As seen, even though a low-rank method, $\ell_{2,1}$-RPCA performs reasonably well but not on par with DPCP-PSGM and REAPER, which overall

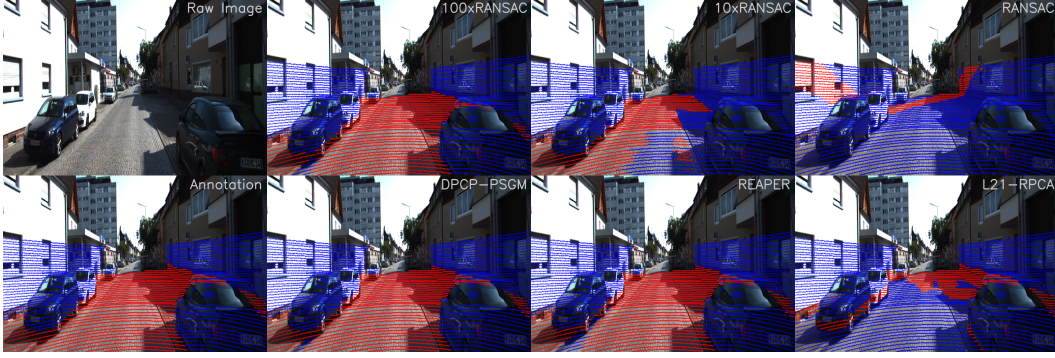

Figure 5: Frame 21 of dataset KITTI-CITY-48: raw image, projection of annotated 3D point cloud onto the image, and detected inliers/outliers using a ground-truth threshold on the distance to the hyperplane for each method. The corresponding F1 measures are DPCP-PSGM (**0.933**), REAPER (0.890), $\ell_{21}$-RPCA (0.248), RANSAC (0.023), 10xRANSAC (0.622), and 100xRANSAC (0.824).

tend to be the most robust methods. On the contrary, for the same time budget, RANSAC, which is a popular choice in the computer vision community for such outlier detection tasks, is essentially failing due to an insufficient number of iterations. Even allowing for a 100 times higher time budget still does not make RANSAC the best method, as it is outperformed by DPCP-PSGM on five out of the seven point clouds (1, 45, and 137 in KITTY-CITY-5, and 0 and 21 in KITTY-CITY-48).

Table 2: Area under ROC for annotated 3D point clouds with index 1, 45, 120, 137, 153 in KITTY-CITY-5 and 0, 21 in KITTY-CITY-48. The number in parenthesis is the percentage of outliers.

| Methods | KITTY-CITY-5 | | | | | KITTY-CITY-48 | |
|---|---|---|---|---|---|---|---|
| | 1(37%) | 45(38%) | 120(53%) | 137(48%) | 153(67%) | 0(56%) | 21(57%) |
| DPCP-PSGM | **0.998** | **0.999** | 0.868 | **1.000** | 0.749 | **0.994** | **0.991** |
| REAPER | 0.998 | 0.998 | 0.839 | 0.999 | 0.749 | **0.994** | 0.982 |
| $\ell_{2,1}$-RPCA | 0.841 | 0.953 | 0.610 | 0.925 | 0.575 | 0.836 | 0.837 |
| RANSAC | 0.596 | 0.592 | 0.569 | 0.551 | 0.521 | 0.534 | 0.531 |
| 10xRANSAC | 0.911 | 0.773 | 0.717 | 0.654 | 0.624 | 0.757 | 0.598 |
| 100xRANSAC | 0.991 | 0.983 | **0.965** | 0.955 | **0.849** | 0.974 | 0.902 |

## 5    Conclusions

We provided an improved analysis for the global optimality of the DPCP method that suggests that DPCP can handle $O((\#\text{inliers})^2)$ outliers. We also presented a scalable first-order method for solving the DPCP problem that only uses matrix-vector multiplications, for which we established global convergence guarantees for various step size selection schemes, regardless of the non-convexity and non-smoothness of the DPCP problem. Finally, experiments on 3D point cloud road data demonstrate that the proposed method is able to outperform RANSAC even when RANSAC is allowed to use 100 times the computational budget of the proposed method. Extensions to allow for corrupted data and multiple subspaces, and further applications in computer vision are the subject of ongoing work.

## Acknowledgment

The co-authors from JHU were supported by NSF grant 1704458. We thank Tianyu Ding of JHU for carefully proof-reading the longer version of this manuscript and catching some mistakes, Yunchen Yang and Tianjiao Ding of ShanghaiTech for refining the 3D point cloud experiment, Ziyu Liu of ShanghaiTech for his help in deriving the concentration inequality for $\eta_{\mathcal{O}}$, and the three anonymous reviewers for their constructive comments.

## Footnotes

[1]This asymptotic result assumes that $d$ and $D$ are fixed, thus these two parameters are omitted.

[2] Table 1 is an adaptation of Table I from [12]. We note that not all bounds are directly comparable because different methods might be analyzed under different models, e.g., for DPCP we use the random spherical model, while for REAPER the random Gaussian model is used. Nevertheless, the two models are closely related, since a random vector distributed according to the standard normal distribution tends to concentrate around the sphere.

[3]Note that if the outliers are in general position, then $\mathcal{H}_1$ will contain at most $D - d - 1$ outliers.

[4]Every $d$-tuple of inliers is linearly independent, and every $D$-tuple of outliers is linearly independent.

[5]This optimization problem also appears in different contexts (e.g., [18] and [21]).

[6]Details of the procedure can be found in the supplementary material, where we are also able to provide an improved analysis for their ALP method.

[7]While smoothing allows one to use gradient-based algorithms with guaranteed convergence, the obtained solution is a perturbed version of the targeted one and thus a rounding step (such as solving a linear program [18]) is required. However, as illustrated in Figure 3, solving one linear program is more expensive than the PSGM for (2) when the data set is relatively large, thus indicating that using a smooth surrogate is not always beneficial.

[8]Coherence Pursuit [19] is not applicable to this experiment because forming the required correlation matrix of the thousands of 3D points is prohibitively expensive.

[9]For RANSAC, we also use each such threshold as its internal thresholding parameter.

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
