[Reviews · NeurIPS 2018]

Reviewer 1



The paper addresses the problem of fitting a subspace to data in cases where the underlying subspace is high-dimensional with respect to the ambient dimension, e.g., a 10-dimensional subspace in 11-dimensional ambient space, in the presence of outliers in data. The problem is also referred to as Dual Principal Component Pursuit (DPCP) in the literature. Recently, [17] has studied the problem, formulating it as finding a hyperplane normal vector $b$ so that for a dataset $X$, the vector $b^T X$ would be sparse, with nonzero elements indicating the outliers. The novelty of the paper is in the presentation of new and stronger theoretical results for the existing algorithm in [17] and a more efficient projected sub-gradient descent method (DPCP-PSGD) to solve the optimization in [17]. The paper demonstrates experiments for road plane detection from 3D data, showing improvement with respect to RANSAC. + The paper is well-written and well-motivated. The review of the existing work in the area is comprehensive. The paper is well-structured and the technical results are explained well. + The theoretical derivations and results in the paper are sound and provide stronger conditions for the success of DPCP, showing that the algorithm can tolerate up to $O(N^2)$ outliers, where $N$ indicates the number of inliers. This improves the existing results that guarantee success with $O(N)$ outliers. + The proposed DPCP-PSGD, while quite standard, for the studied problem achieves improved computational time with respect to existing solvers. The paper also proves linear convergence for the proposed algorithm. - Given that the setting of the experiment in the paper is to find 2D planes in 3D, to understand the effectiveness of the proposed method with respect to the state of the art, it is necessary that the paper provides comparison with more compelling baselines and state of the art other than RANSAC. In particular, the reviewer suggests comparing with "Robust PCA via Outlier Pursuit, H. Xu, C. Caramanis, S. Sanghavi" as well as [13] and [17], which all address the problem of handling outliers when fitting a subspace to data.

Reviewer 2



This work has provided an improved analysis for global optimality of the (nonconvex) dual PCA problem based on random model assumption, and a more efficient optimization method based sub-gradient descent with a local linear convergent rate. The paper is well-organized overall, and the theoretical improvement is substantial compared to the existing work. The explanations of analysis in Section 2 and algorithms in Section 3 are not very clear and can be improved in the future. Below are some more detailed comments: 1. Assumptions of the random model: First, aside from the benefits of the analysis, how does the random spherical model of the subspace relates to applications? Does real data (both outliers and subspace) follow random spherical distribution? The reviewer feels that more validation is needed. Second, the description of the random model of the subspace in Theorem 2 is not very clear. From the reviewer’s understanding, is it that the columns of O is uniformly drawn from the sphere S^{D-1}, and the columns of X is uniformly drawn from a lower dimension sphere that S^{D-1} \cap A, where A is and dimensional subspace? The current description is not very clear and quite misleading. Besides, is there any assumption for the properties of the subspace A? 2. The relationship between M and N. From Theorem 2, if the reviewer understand correctly, the authors hid the dimension D and d in the relation of M and N. If the dimension D and the subspace dimension d are counted, it should be M<=O(d/D*N^2), which makes sense as the subspace dimension d (ambient dimension D) gets bigger, more (fewer) outliers are tolerated. As the dimension D could be very large for many applications, the reviewer feels those parameters (d and D) cannot be ignored and should be accounted and explained in the introduction and the remarks. 3. For the subgradient method, could the author explain the intuition behind the choice of the initialization? Why does such a choice produce an initialization with small principle angle? On the other hand, for the subgradient method, does the linear convergence of subgradient method mainly due to the choice of the step size or due to the structure of the problem? For general convex nonsmooth problems, the convergence rate of subgradient method is usually sublinear. The reviewer feels that the reason could be that the local region around the optimal solutions has certain benign geometric structures (i.e. satisfies certain regularization conditions) that make linear convergence possible. It would be better if the author provides better explanation of this surprising’’ algorithmic property. 4. Possible Extensions: (1) The current work only considers finding a single subspace, does the current analysis possible to be extended to multiple (orthogonal) subspaces (e.g. using deflation or optimization over oblique manifold)? (2) Instead of l1 norm loss, the author could consider a Huber loss as analyzed in [21], which is first order smooth which is much easier to analyze. 5. typos: Riemann gradient => Riemannian gradient

Reviewer 3



The rebuttal addressed a few minor confusions I had. ------ This paper considers the problem of subspace learning, in presence of nontrivial portion of outliers. Especially, this paper develops improved theory and algorithm that O(n^2) outliers can be tolerated, rather then O(n). The paper is clear and well written. The paper characterize the geometric property as either orthogonal or close to the desired subspace in Lemma 1, and then derive the necessary scaling for any global solution to be orthogonal to the subspace. Algorithmically, initialized on a point, a natural gradient descent procedure can recover a good approximation of the normal direction. Performance guarantee wrt different step sizes are also discussed. * In figure 2, the curve for tan(theta_k) is not monotone, could the authors give more intuition why this is the case? * Could the author give some intuition why the geometrically diminishing step size matters for this problem or other general nonconvex problem? * In figure 3, the proposed algorithm seems to take more time and iterations compared to ALP. This does not support the argument made earlier about computation efficiency. Maybe by increasing the size of the problem, this point could be made more clear.